# Development of Debiasing Technique for Lung Nodule Chest X-ray Datasets to Generalize Deep Learning Models

**DOI:** 10.3390/s23146585

**Published:** 2023-07-21

**Authors:** Michael J. Horry, Subrata Chakraborty, Biswajeet Pradhan, Manoranjan Paul, Jing Zhu, Hui Wen Loh, Prabal Datta Barua, U. Rajendra Acharya

**Affiliations:** 1Centre for Advanced Modelling and Geospatial Information Systems (CAMGIS), Faculty of Engineering and Information Technology, University of Technology Sydney, Ultimo, NSW 2007, Australia; michael.j.horry@student.uts.edu.au (M.J.H.); biswajeet.pradhan@uts.edu.au (B.P.); prabal.barua@usq.edu.au (P.D.B.); 2IBM Australia Limited, Sydney, NSW 2000, Australia; 3Faculty of Science, Agriculture, Business and Law, University of New England, Armidale, NSW 2351, Australia; 4Earth Observation Center, Institute of Climate Change, Universiti Kebangsaan Malaysia, Bangi 43600, Malaysia; 5Machine Vision and Digital Health (MaViDH), School of Computing and Mathematics, Charles Sturt University, Bathurst, NSW 2795, Australia; mpaul@csu.edu.au; 6Department of Radiology, Westmead Hospital, Westmead, NSW 2145, Australia; zhujingzhou@gmail.com; 7School of Science and Technology, Singapore University of Social Sciences, Singapore 599494, Singapore; hwloh002@suss.edu.sg; 8Cogninet Brain Team, Cogninet Australia, Sydney, NSW 2010, Australia; 9School of Business (Information Systems), Faculty of Business, Education, Law & Arts, University of Southern Queensland, Toowoomba, QLD 4350, Australia; 10School of Mathematics, Physics and Computing, University of Southern Queensland, Springfield, QLD 4300, Australia; rajendra.acharya@usq.edu.au

**Keywords:** chest X-ray, confounding bias, deep learning, model generalization, lung cancer, federated learning

## Abstract

Screening programs for early lung cancer diagnosis are uncommon, primarily due to the challenge of reaching at-risk patients located in rural areas far from medical facilities. To overcome this obstacle, a comprehensive approach is needed that combines mobility, low cost, speed, accuracy, and privacy. One potential solution lies in combining the chest X-ray imaging mode with federated deep learning, ensuring that no single data source can bias the model adversely. This study presents a pre-processing pipeline designed to debias chest X-ray images, thereby enhancing internal classification and external generalization. The pipeline employs a pruning mechanism to train a deep learning model for nodule detection, utilizing the most informative images from a publicly available lung nodule X-ray dataset. Histogram equalization is used to remove systematic differences in image brightness and contrast. Model training is then performed using combinations of lung field segmentation, close cropping, and rib/bone suppression. The resulting deep learning models, generated through this pre-processing pipeline, demonstrate successful generalization on an independent lung nodule dataset. By eliminating confounding variables in chest X-ray images and suppressing signal noise from the bone structures, the proposed deep learning lung nodule detection algorithm achieves an external generalization accuracy of 89%. This approach paves the way for the development of a low-cost and accessible deep learning-based clinical system for lung cancer screening.

## 1. Introduction

Lung cancer is the leading cause of cancer mortality worldwide with 1.80 million deaths documented by the World Health Organization in 2020 [1]. The global deaths worldwide attributable to lung cancer are twice those from the second most common cause of cancer death, colorectal cancer. Studies have shown a favorable prognosis for early-stage lung cancer, with five-year survival rates of up to 70% for patients with small, localized tumors [2]. Relieving the economic and sociological costs associated with lung cancer is therefore dependent on the early diagnosis of this condition.

A comprehensive meta-review [3] concluded that low-dose computed tomography (LDCT) lung cancer screening could reduce lung cancer-associated mortality by 17% and overall mortality by 4% in risk populations highly vulnerable due to tobacco exposure. Although LDCT is more effective at detecting lung cancer at an early stage than chest X-ray (CXR) [4], the feasibility and cost-effectiveness of LDCT screening in low- and middle-income countries without well-developed healthcare infrastructures have been questioned [5]. In some rural areas lacking transportation, long distances and poor road conditions can make healthcare inaccessible for many [6]. Compounding this problem is the relative lack of skilled radiologists and oncology professionals per person in the population, particularly in low- and middle-income countries. Furthermore, even in high-income countries, LDCT is known for its tendency to produce false positives, resulting in invasive procedures to characterize the nodule and associated false positive nodule workup risks [7]. For these reasons, lung cancer screening remains uncommon, with only the USA and China currently implementing lung cancer screening. These programs target high-risk lifetime smokers, in contrast to the entire population screening programs common for other cancers, such as breast, cervical, and colorectal cancers [8].

Extending the scope of lung cancer screening to cover a broader population depends on increasing community access to screening facilities at affordable costs. CXR is a widely available, safe, simple to operate, and inexpensive medical imaging technology compared to LDCT [9]. In addition, the CXR image acquisition apparatus is readily available in a portable/mobile form that is easily cleaned and maintained. For these reasons, CXR remains a potentially appealing technology for lung cancer screening programs, especially in situations in which clinical resources are limited, and patients are located far from health infrastructure [10].

Implementation of the CXR imaging mode and deep learning techniques to boost radiologist productivity and sensitivity could, in turn, provide a pragmatic and economically feasible mechanism for broad population screening for lung cancer, potentially saving many lives and reducing the economic impact of this disease. A population-wide lung cancer CXR screening program would provide a valuable corpus of training data for lung cancer detection deep learning models but only if the collected images were free of confounding variables and bias, with privacy guaranteed. Much has been published in relation to the use of federated learning as a technology to implement privacy by design [11,12], with one study [13] investigating the susceptibility of these systems to biases caused by adversarial images. There has been little investigation into the effects of homogenizing medical image data to remove biases and thereby improve model generalization. This study proposes an image pre-processing pipeline that simultaneously homogenizes and debiases chest X-ray images, leading to improvements in internal classification and external generalization and paving the way for federated approaches for effective deep learning lung nodule detection. This study is the first that systematically assesses the utility of combining several debiasing techniques using a process of ablation to determine the impact of each.

### 1.1. Related Work

Although many studies have used deep learning algorithms to classify thoracic disease [14,15,16,17,18,19,20,21,22,23], very few have applied image homogenization techniques apart from the near-universal application of histogram equalization. Lung field segmentation was used by [24] to improve both areas under the curve (AUC) and disease localization for 14 thoracic conditions from the chest X-ray14 dataset [23], but no external generalization testing was performed in their study. Lung field segmentation was also used in a study by [25] as a pre-processing step in CXR-based nodule localization and characterization, leading to state-of-the-art results for this task. However, ablation testing excluding the segmentation algorithm was not performed. A small number of studies have considered the effect of rib and bone suppression on automated lung nodule detection [26,27,28]. Of these studies, [29] is the most recent and comprehensive, being an investigation of deep learning with lung field segmentation and bone suppression to improve automated nodule detection for the JSRT dataset. This study found a segmented and bone-suppressed image corpus led to better deep learning CNN training and validation accuracy when combined with the exclusion of outlying records (5% of all images). External validation of these results was not reported.

An alternative line of research into debiasing deep learning CXR automated diagnosis systems is the use of custom loss functions to disentangle features, thereby achieving confounder-free training [30,31]. The idea involves including the training data source as a “bias” feature that penalizes gradient descent, with that feature allowing the model to distinguish the source dataset provenance. In the context of COVID-19 detection [30], this approach improved the AUC by 13% on held-out data, outperforming both histogram equalization and lung field segmentation as debiasing techniques. To date, the most advanced form of a fair confounder-free training algorithm can be found in [31], allowing for a continuously variable confounder value, rather than the binary or discrete values typically found with disentanglement training algorithms [32,33,34]. This study successfully applied confounder-free learning to several tasks, including determining bone age from pediatric hand X-ray images and reducing the effect of sex as a confounding variable. However, the current state of the art is limited by a lack of advanced knowledge of the confounding variable to be suppressed using this technique and the creation of inappropriately conditioned datasets [31].

This study aims to address bias and confounding at the source, being the medical images themselves since, in the federated learning context, data sources are indeterminate. Therefore, source-based confounding variables would be too diverse to be mitigated by feature disentanglement approaches alone.

### 1.2. Importance to the Field

The recent worldwide COVID-19 pandemic provided the research community with a unique opportunity to validate computer vision-based CXR analysis techniques in a real-world application since the accessibility of CXR as an imaging mode made it an ideal tool for tracking COVID-19 lung involvement progression. Despite promising early results, many development studies proved to be overly optimistic due to confounding bias, whereby machine learning-based classifiers chose “shortcuts” over signals especially when disease-positive and control sample images were independently sourced [35]. In such cases, source-dependent systematic differences in image attributes, such as brightness, contrast, labeling, projection, and patient position, could overwhelm the pathological signal differences between the classes, resulting in unreliable classification results for real-world clinical applications. Confounding variables can also come from demographic factors, such as the sex, socioeconomic status, and age of sample populations [36]. Egregious examples of confounded machine learning-based CXR studies are the numerous COVID-19 diagnostic studies that have reported extremely high diagnostic accuracy for COVID-19 pneumonia against “other” viral pneumonias without recognizing that the COVID-19 sample population consisted of elderly patients, while the viral pneumonia control population consisted of pediatric samples, as reported by [37].

The problem of confounding bias is a core inhibitor of the clinical acceptance of machine learning CXR analysis algorithms. The usefulness of such systems depends upon the quantity and quality of the labeled training data upon which they are trained. Systematic differences caused by demographic factors, image acquisition apparatus calibration and operation, projection, and regional morphology diversity will always be present. It follows that biases in training data corpora are inherently unavoidable. Therefore, a practical and reliable approach to CXR homogenization is needed to allow deep learning models to generalize unseen datasets.

## 2. Materials and Methods

Deep learning models were developed in Python/Tensorflow [38] in the University of Technology Sydney interactive high-performance computing environment.

### 2.1. Datasets

A process of bibliographic analysis was undertaken to determine the list of publicly available datasets containing CXR images labeled with lung cancer or conditions associated with lung cancer. To identify lung cancer-specific datasets, a title/abstract/key search was performed in Scopus using the following query string:

(TITLE-ABS-KEY (chest AND (X-ray or radiograph) AND (lung or pulmonary) and (nodule or mass or tumor) AND (dataset or database))).

This query returned 451 results. A document-based citation analysis was performed using VOSViewer [39], whereby the relatedness of the items was based simply on the number of times that items had cited each other. The reasoning was that the publicly available data sources would be among the most highly cited items in this field of research. Of the 451 items, the largest set of connected (by citation) items was 153, resulting in a graph consisting of seven clusters. Each paper in each cluster was then retrieved for full text review to determine the underlying dataset used for the study. In addition, common themes within the cluster were identified and logged in Table 1.

It was clear from this analysis that the most widely studied lung nodule dataset is the Japanese Society of Radiological Technology database (JSRT) [40] with the CXR subset of the Lung Image Database Consortium Image Collection (LIDC) [51], along with the Prostate, Lung, Colorectal, and Ovarian (PLCO) [41] dataset also used in influential studies. The JSRT dataset is the most cited data source with 944 citations at the time of writing. The use of this dataset across numerous studies has resulted in it being the de-facto standard dataset for comparative studies of automated lung nodule detection. For this reason, we chose the JSRT dataset as the CXR corpus for training deep learning models in this study. The JSRT dataset also provides gold-standard lung field masks that allow for good-quality segmentation of the dataset and training data for our U-Net [52]-based lung segmentation algorithm, discussed in Section 2.3. Finally, the JSRT authors conducted an AUC-ROC study using 20 radiologists from four institutions determining an average AUC of 0.833 ± 0.045. This knowledge was useful for calibrating our expectations for model performance and for sanity-checking our internal testing results. We selected a stable AUC of 80% as the target reference to compare the number of images pruned for successful internal training of each model.

For external testing, we used the CXR subset of the LIDC. This dataset includes 280 CXR images with expert labels indicating the presence and malignancy of lung cancer. Four radiologists labeled this dataset with access to the corresponding patient computed tomography (CT) scans to confirm the lung cancer diagnosis. This dataset was used as an inference-only dataset to check the generalization of the JSRT-trained models with various configurations of the pre-processing pipeline. Additional properties of the JSRT and LIDC datasets are provided in Table 2.

### 2.2. Image Pre-Processing Operations

#### 2.2.1. Histogram Equalization

CXR images exhibit variable contrast based on the technical calibration of the image acquisition apparatus, particularly, the energy of the primary beam and the application of scatter radiation minimization methods, such as collimation, grids, or air gaps [54]. Histogram equalization processing has proven to be effective in normalizing the gray-level distribution of CXR images, resulting in uniform image histograms [55]. Histogram equalization was applied to all images in this study as a standard first step toward image homogenization. This step is required since deep learning-based computer vision algorithms are trained on continuous pixel intensity values, and systematic differences in image contrast between the datasets would result in biased model training and negatively impact model generalization.

#### 2.2.2. Lung Field Segmentation

Lung nodules occur within the lung field; therefore, non-lung field pixels are signal noise. CXR features, such as projection labels and portability markers at the edges of CXR images, have been shown by several studies to be sources of confounding bias in deep learning-based medical imaging applications [35,56,57]. Projection labels are effective predictors of CXR repository sources with accuracy of up to 99.98% and 100% accuracy in separating emergency room from inpatient CXR images [56]. In addition, the use of bedside mobile CXR apparatus is frequently associated with more severe disease (due to lack of patient mobility), allowing deep learning models to use apparatus portability labels as a signal indicating the disease and its severity. Thus, eliminating non-lung field pixels using segmentation effectively removes training bias and results in more generalizable deep learning models.

The utility of this approach in medical image classification has been supported in previous studies [58,59]. A comprehensive review of lung area segmentation techniques may be found in [60]. The best performance reported by this survey was achieved using a CNN-based deep learning system with a Dice similarity coefficient of 0.980 [61]. A much simpler approach leveraging the U-Net architecture [52] was trained on the JSRT dataset [40] consisting of 385 CXR images with gold-standard masks to achieve a Dice similarity co-efficient of 0.974 [62]. We reasoned that the U-Net-based architecture could be improved in terms of Dice similarity coefficients, as well as adaptability, using training data from two additional data sources, the Montgomery and Shenzen datasets [63], thereby creating a combined training data corpus of 1185 CXR image/mask pairs.

Lung field segmentation experiments were performed using a VGG-like CNN [64], DenseNet [65], and ResNet [66]-based U-Net architectures. We found that a Deep Residual U-Net following [67] provided the highest Dice similarity scores and was consistent with other studies [68,69] outside the medical imaging domain. Since nodules are typically small, we expanded the image size for the residual U-Net implementation from 224 × 224 pixels as used in the original study [67] to 512 × 512 pixels by the addition of extra encoder/decoder residual blocks to maximize the image size with reasonable processing time in our computing environment. This process allows the segmented images to retain important anatomical features for downstream classification.

This combination of training datasets and network design achieved a maximum validation Dice similarity co-efficient of 0.988 at epoch 93, which is a small improvement from [70], who reported a result of 0.974 for lung field segmentation using the JSRT, Shenzen, and Montgomery datasets, and [62], who used the JSRT dataset. Training loss, accuracy curves, and Dice charts for our residual U-Net are included in Figure 1.

When applied to the unseen LIDC dataset, the segmentation network produced minimal artifacts, similar to those reported in [71] on six of the 280 source images, as shown in Figure 2. These artifacts took the form of pinholes in the generated lung mask. These pinholes were eliminated using morphological closing with a kernel size of 8, followed by flood fill of any contour smaller than a parameterized minimum square area, which was determined by the experiment to be optimal at 1/16th of the image area. The before and after results of the parameterized morphological closing/contour filling operations are shown in Figure 2.

A montage of LIDC lung field segmentation masks is shown in Figure 3. Four image masks (highlighted in red) were considered failures for either having a contour count of fewer than 2 or poor visibility of a single lung field. The images associated with these masks were not segmented. Instead, they were removed from the LIDC dataset for generalization testing experiments.

#### 2.2.3. Segmented Lung Field Cropping

Following the segmentation of the CXR images using gold standard masks for JSRT and U-Net generated masks for LIDC, the size of the lung field on the images was observed to be highly variable due to differences in patient morphology and main beam focal distance, as shown in Figure 4a. It may be noted that the area surrounding the lung field contains no useful information and could potentially be a source of confounding bias. Therefore, the CXR images were closely cropped on all sides to ensure that the first non-black pixel is on the border of the image for all directions, per Figure 4b.

#### 2.2.4. Rib and Bone Suppression

We noted that the ribcage features are very pronounced on the CXR images after equalization and segmentation. Additionally, ribs in the LIDC dataset generally appeared to be more prominent than ribs in the JSRT dataset. Therefore, we considered the appearance of ribs as a potential confounding variable and sought to suppress and homogenize the appearance of ribs on the CXR images. Two approaches were investigated: first, an autoencoder-based suppression algorithm was trained using bone-suppressed JSRT images [72]; and second, we adopted a simple CNN-based approach following [73] by reusing code provided to the community [74]. We found that the second approach resulted in much sharper images with less blurring of the fine lung structure details, consistent with the results of [73]. The before and after rib suppression, for example, CXR images, are presented in Figure 5a–d.

### 2.3. Deep Learning Model

The JSRT image dataset was loaded into data/label arrays and randomly split into four parts using the scikit-learn StratifiedKFold function [75]. A VGG16 [64] classifier initialized with ImageNet [76] weights was selected following experimentation with VGG16, VGG19 [64], DenseNet-121 [65], and ResNet50 [66]. The pre-trained VGG16 model showed the least tendency to overfit and provided the most uniform training results for each part of the JSRT dataset, consistent with our earlier results on small medical image datasets with insufficient data to train deeper networks [77].

The pre-trained VGG16 classifier was modified with a hand-crafted output head designed to regularize the network and facilitate binary classification of the nodule and non-nodule image classes (see Figure 6).

The modified VGG16 classifier was trained for each of the four parts using a binary cross entropy loss function with a learning rate of 0.0005 for 50 epochs and 10 fine-tuning epochs at a lower learning rate of 0.00001 to avoid overfitting. The trained models were captured at the point of minimum validation loss. Sample training and fine-tuning accuracy/loss curves are shown in Figure 7.

### 2.4. Evolutionary Pruning Algorithm

During initial four-fold training/validation, we observed great variation in the validation accuracy of per-fold models despite the model and training hyperparameters being held constant. This variability was due to the randomized images per fold producing some dataset shards that trained more easily than others.

To identify the nodule image samples that were the least informative for the nodule class, an inference operation was performed against each nodule CXR image in the JSRT dataset following the completion of each training round. This inference operation tested all four shards against the k-fold trained model, resulting in 16 inferences per nodule positive image (4 data shards × 4 models). For each round, the JSRT nodule image with the largest number of misclassifications was added to a prune list and ignored on the next training round, thereby under-sampling the majority nodule image class. This process was repeated 61 times, and the number of nodule-positive images was reduced from 154 to 93 to balance the non-nodule image sample number. The pruned JSRT dataset was then used to train a master model for external generalization testing against the LIDC dataset. This end-to-end process of pruning and external testing is illustrated in Figure 8.

### 2.5. Experiment Setup and Ablation Studies

This study was concerned with any potential effects from the mix-and-match combinations of lung field segmentation, close cropping, and rib suppression on internal and external classification results. Image histogram equalization was applied in all experiments as a standard image processing technique that was implemented in line with data ingestion.

This process resulted in six experiment configurations, as detailed in Table 3. It may be noted that the cropping operation was only applied in combination with the segmentation operation (experiments E and F). Experiments A, C, and E did not include rib suppression, while experiments B, D, and F employed rib suppression. Experiments C and D applied lung field segmentation as a proposed debiasing process, and experiments E and F further applied cropping to the segmented lung field. Each experiment may be considered an ablation study for the complete debiasing pipeline represented by experiment F.

## 3. Results

### 3.1. Internal Four-Fold Training and Testing

The internal testing results for experiments A–F are illustrated in Figure 9a–f using a plot showing the number of pruned images on the *x*-axis and the corresponding calculated AUC values on the *y*-axis.

#### 3.1.1. Internal Testing Result in A (No Debiasing Operations)

Internal testing results for experiment A, which represent training with raw data, are presented in Figure 9a. The initial training round achieved an AUC of around 67.6% ± 4.1%, consistent with [23] and another deep learning study using the JSRT dataset [78]. However, these results improved significantly as the poorest performing nodule test samples were pruned, achieving a stable 80% with 15 pruned records and 93.5% ± 2.4 when the classes were balanced with 62 pruned records.

#### 3.1.2. Internal Testing Result B (Rib Suppression Operator)

Internal testing results following the application of the rib suppression operator only are shown in Figure 9b. Rib suppression resulted in a significant improvement of the initial results to 70.4 ± 5.0%. Once again, the results improved linearly as the poorest test nodule samples were pruned, with a stable 80% AUC achieved at around 15 pruned records with an AUC of 92.6 ± 4.6% achieved at the point of class balance.

#### 3.1.3. Internal Testing Result C (Segmentation Operator)

Application of the lung field segmentation operator resulted in a further improvement of the initial internal testing results, with an AUC of 72.6 ± 1.6% achieved without record pruning, as shown in Figure 9c. The results improved following pruning of the poorest performing lung nodule test images. However, this improvement occurred at a lower rate than in both experiments A and B. A stable AUC of 80% was not reached until around 35 pruned records. This slower rate of AUC improvement per pruned image could be interpreted as improved classification sensitivity for the more challenging nodule images. Despite the improved initial AUC, the slower rate of improvement resulted in a lower AUC of 88.1 ± 1.0 at the point of class balance. The lower final AUC and a very low error margin may have been caused by the removal of confounding variables outside the lung field that allowed models A and B to take shortcuts in learning.

#### 3.1.4. Internal Testing Result D (Segmentation + Rib Suppression Operators)

Application of lung field segmentation and rib suppression to the histogram equalized images resulted in another incremental improvement in the initial testing round with an AUC of 74.9 ± 3.9% with no pruned records, as shown in Figure 9d. A stable AUC of 80% was reached after pruning only 13 records, representing a significant improvement over rib suppression alone (experiment C). An AUC of 90.3 ± 4.1% was achieved at the point of class balance, again a significant improvement over segmentation alone.

#### 3.1.5. Internal Testing Result E (Segmentation + Cropping Operators)

Compared to segmentation only, the cropping operation reduced the initial AUC to 70.7 ± 11.4% but allowed the model to reach a stable 80% AUC at around 25 pruned records (compared to 35 for segmentation only), as shown in Figure 9e. An AUC of 91.0 ± 2.4 was achieved at the point of class balance with 62 pruned records, which is much higher than the final AUC of 88.1 ± 1.0 achieved using segmentation alone.

#### 3.1.6. Internal Testing Result F (Segmentation + Cropping + Suppression Operators)

The best result was achieved in experiment F using all operators, as shown in Figure 9f. The initial average AUC was 74.2 ± 7.1%. However, this experiment achieved a stable 80% AUC from 10 pruned records with an AUC of 90.5 ± 4% achieved at the class balance point of 62 pruned nodule images. Although other experiments may have achieved higher AUC values, this outcome was only possible after discarding far more difficult nodule images. Experiment F provided the best balance of excellent initial results, with minimal pruning of only 10 of 293 or 3.4% of nodule images to reach the target stable 80% AUC.

### 3.2. Pruned Records Analysis (from Experiment F)

Since experiment F achieved excellent results with minimally pruned nodule records, we were interested in the attributes of the pruned records potentially leading our trained CNN to misclassify the nodule images as non-nodules. Therefore, we consulted a practicing radiology registrar to interpret the top five misclassified JSRT nodule images, as shown in Table 4. Each of the images that were found to be challenging to automatically classify was also challenging for human readers due to low visibility thresholds, such as nodule features hidden behind the cardiac silhouette and/or position overlapping bone, vascular marking, or breast tissue.

### 3.3. External Testing Evaluation Using Pruned Models

A series of master models were trained for each experiment using JSRT-based datasets that had been balanced by pruning the over-represented nodule classes, keeping only the images that were most informative of the nodule classification by testing over four folds, as described above. This resulted in six fully trained models used to infer nodule scores for each image in the LIDC CXR subset.

All CXR images in the LIDC dataset are lung cancer cases, with nodules identified by CT scan, which are sometimes very subtle or invisible to human readers on CXR images. A final experiment was conducted to identify which image processing operators had promoted the generalization of the JSRT-trained models to an independent dataset. The results are presented in Figure 10, in which the nodule class probability has been plotted for each LIDC image in a scatterplot.

Since all LIDC images are of lung cancer patients, the plot can be expected to have most data points at greater than 0.5 probability if the JSRT model has been successfully generalized to the LIDC dataset. This is the case for models implementing the rib-suppression operator (experiments B, D, and F), particularly those in which the lung field segmentation operator was also used (experiments D and F), with the best results achieved in experiment F using the combination of rib suppression, lung field segmentation, and close cropping operators. This combination of operators achieved a classification accuracy of 89% with LIDC nodule images which we considered to be an excellent result given the subtlety of nodule features on some of the LIDC images; some nodules were measured to be smaller than 3 mm on CT scans [51], and smaller nodules were poorly visible on the corresponding CXR images.

## 4. Discussion

These experiments delivered promising results. The first key observation is that image histogram equalization alone, with no further debiasing operators, resulted in models that did not generalize at all, per Figure 10a. This fact is a concern given that most studies implementing deep learning in automated analysis of CXR images used histogram equalization as the only pre-processing step. The second interesting result is that lung field segmentation alone worsened the classification metrics internally and externally. We see this outcome as strong evidence of confounding variables present in the JSRT image data corpus that allowed some degree of shortcut learning from unsegmented images, which was eliminated when these images were segmented. Finally, the poor external testing result provided evidence that lung field segmentation alone is not enough to sufficiently debias an image corpus to promote generalization.

Our best generalization results were achieved using the rib suppression operator. Although this technique has been shown to improve the sensitivity of human radiologists to nodules [79], there has been little published literature using rib and bone suppression in deep learning model generalization for medical images. This study is the first to employ this method to the best of our knowledge. In our work, lung field segmentation with and without close cropping improved the external classification results, but only after applying the rib suppression operator. This outcome shows that the appearance of ribs on CXR images is a confounding variable, which matched our observations that the visibility of ribs on CXR images significantly varies between datasets. Improvement in generalization only becomes evident after lung field segmentation and cropping, when this confounder is suppressed.

### Limitations

The technique developed in this study could only detect nodules within the lung field. It cannot detect nodules occurring in the heart, mediastinal, and diaphragm regions because the lung field segmentation process masks these areas. It is important for radiologists to carefully examine other anatomical regions within the pleural cavity that are prone to being overlooked, such as vascular features within the hilum, apices, and rib-crossings [80]. Computer vision algorithms, including the one developed in this study, are also expected to face challenges in distinguishing these features from lung nodules.

This study was constrained by the datasets publicly available from JSRT and LIDC. The JSRT dataset does not include images with medical devices, such as pacemakers and drains; surgical modifications, such as resection with displacement; or significant skeletal deformities, such as kyphosis. For a clinically useful model, it would be necessary to train using cases that encompass these feature classes. This aspect will be addressed in future research endeavors. Considering that privacy concerns restrict access to CXR data, the future progress of this study will involve adoption of a federated learning paradigm in conjunction with a clinical study. This approach will allow for more extensive access to real-world training data and enable clinical validation of the proposed debiasing pre-processing pipeline. The ultimate aim is to develop a generalized lung nodule detection reference implementation.

## 5. Conclusions

The potential of private, federated deep learning to improve access to training data while maintaining privacy can only be realized with an effective debiasing image pre-processing pipeline. In this study, we demonstrated that the combination of histogram equalization, rib suppression, and close-cropped lung field segmentation effectively homogenized and debiased a corpus of CXR images, enabling trained models to generalize to external datasets that utilize the same image pre-processing pipeline. Notably, this study is the first to incorporate a rib suppression operator in an external generalization study. The inclusion of this operator was essential in achieving external generalization accuracy of 89% for a model trained on the JSRT dataset and evaluated against the external LIDC dataset. This state-of-the-art result should instill confidence in the research community that deep learning classifiers can be trained in a bias-free manner, enabling their application across datasets from different sources. This advancement paves the way for the development of valuable clinical tools with broad applicability. We envision that such a tool, utilizing the CXR imaging mode, can facilitate lower-cost and lower-risk lung cancer screening, leading to improved access to early diagnosis and potentially saving many lives while reducing the societal burden associated with this deadly disease.

## Figures and Tables

**Figure 1 sensors-23-06585-f001:**
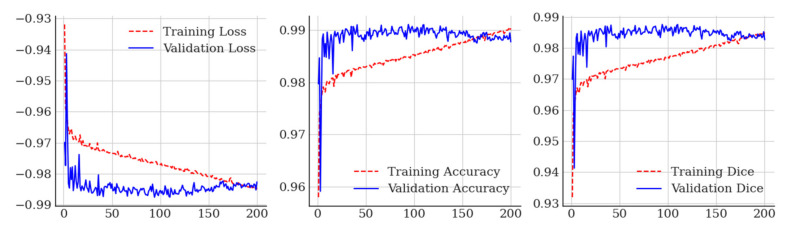
Training curves obtained for deep residual U-Net lung field segmentation.

**Figure 2 sensors-23-06585-f002:**
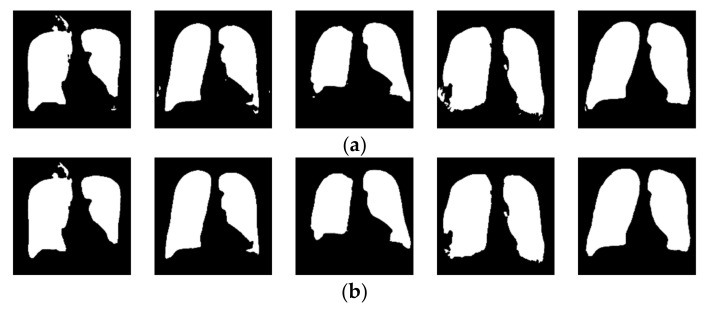
Results of morphological closing and parameterized contour filling to complete lung masks for LIDC images. (**a**) Before the closing of pinhole artifacts; (**b**) after the closing of pinhole artifacts.

**Figure 3 sensors-23-06585-f003:**
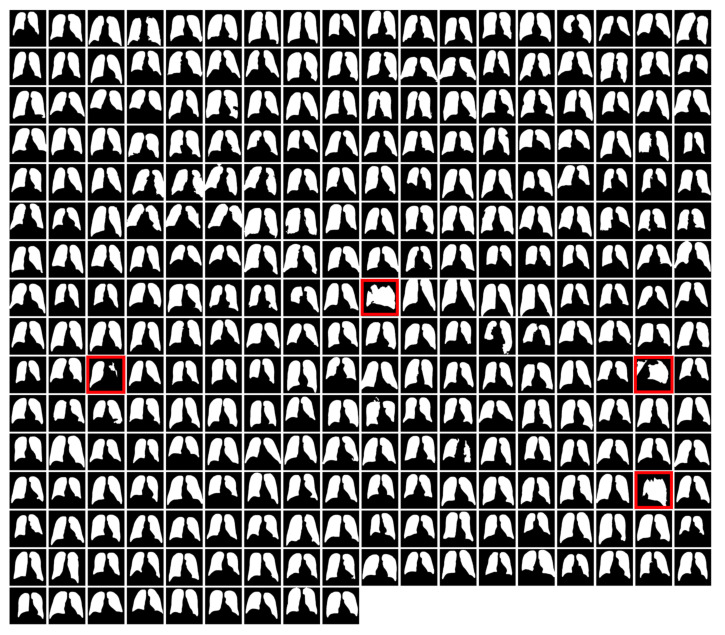
LIDC lung field masks were automatically generated using a hybrid U-Net and single-parameter morphological algorithm.

**Figure 4 sensors-23-06585-f004:**
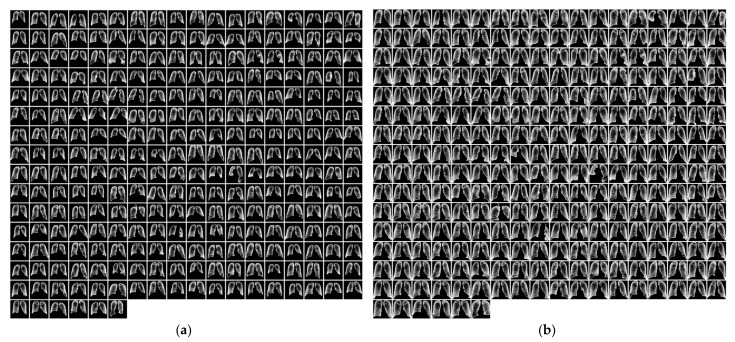
(**a**) Segmented JSRT images; (**b**) segmented and cropped JSRT images. JSRT lung field masks before cropping (**a**) and after cropping (**b**).

**Figure 5 sensors-23-06585-f005:**
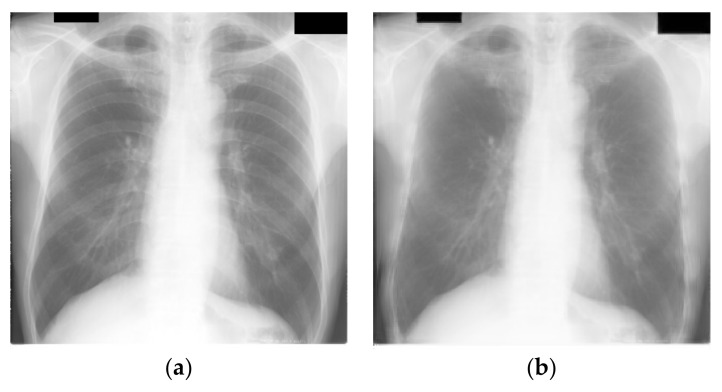
Effect of rib suppression for the JSRT and LIDC datasets. Fine vascular detail has been preserved well with minimal overall reduction in image sharpness. (**a**) JSRT example without rib suppression; (**b**) JSRT example with rib suppression; (**c**) LIDC example without rib suppression; (**d**) LIDC example with rib suppression.

**Figure 6 sensors-23-06585-f006:**
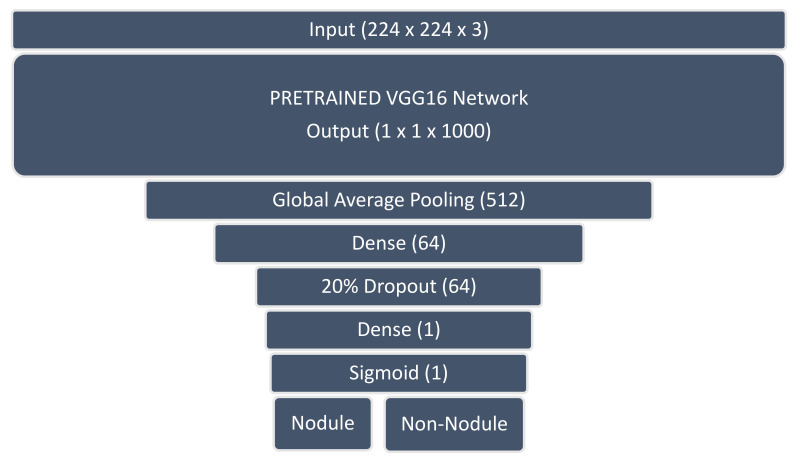
Pre-trained VGG16 classifier with modified output head using sigmoid activation for binary classification.

**Figure 7 sensors-23-06585-f007:**
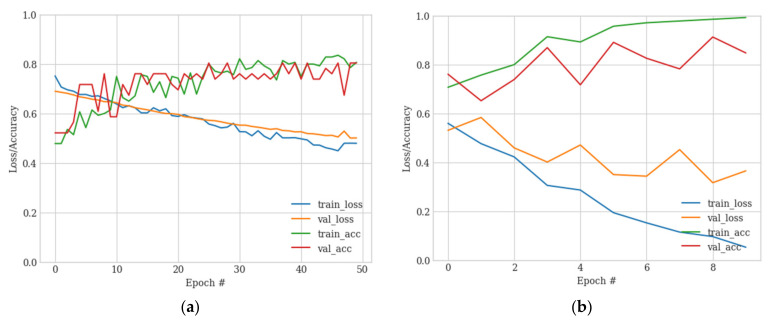
VGG16 training/loss curves for base training (**a**) and fine-tuning (**b**). (**a**) Initial training for 50 epochs; (**b**) fine-tuning for 10 epochs.

**Figure 8 sensors-23-06585-f008:**
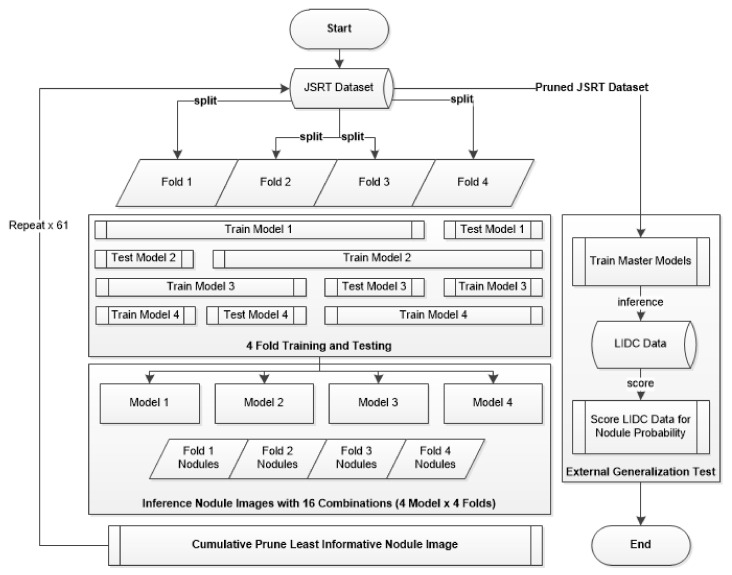
Four-fold training/pruning and external testing flowchart.

**Figure 9 sensors-23-06585-f009:**
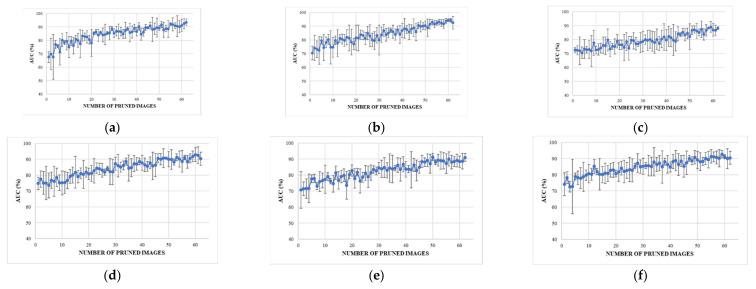
The results obtained from experiments are presented as (**a**–**f**) respectively. (**a**) represents the results obtained from the raw data. (**b**) includes histogram equalization and rib suppression. (**c**) includes histogram equalization and lung field segmentation. (**d**) includes histogram equalization, rib suppression, and lung field segmentation. (**e**) includes histogram equalization, lung field segmentation, and cropping to the lung boundary. (**f**) includes histogram equalization, rib suppression, lung field segmentation, and cropping to the lung boundary.

**Figure 10 sensors-23-06585-f010:**
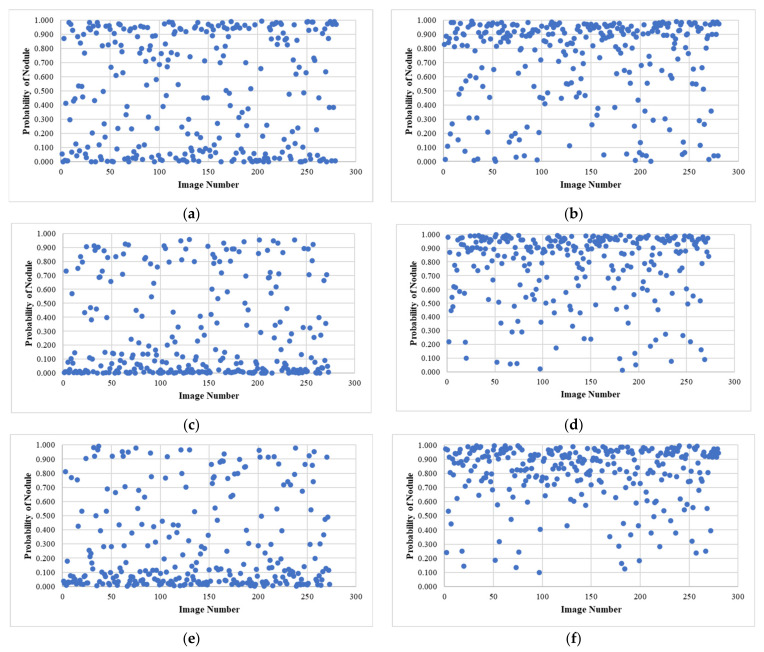
Results of external testing against the LIDC dataset using JSRT-trained models with all combinations of proposed debiasing operators showing the effectiveness of rib suppression in improving classification results (**b**,**d**,**f**) especially when lung field segmentation is also applied (**d**,**f**). (**a**) Experiment A—no debiasing operations; (**b**) Experiment B—rib suppression only; (**c**) Experiment C—segmentation only; (**d**) Experiment D—segmentation and rib suppression; (**e**) Experiment E—segmentation and cropping; (**f**) Experiment F—segmentation, cropping and rib suppression.

**Table 1 sensors-23-06585-t001:** Description of published research clusters.

Cluster	Number of Papers	Central Paper/s	Imaging Mode	Connecting Theme	Data Source
1	36	[40,41]	CXR	Connected by the use of JSRT [40] and PLCO [41] datasets as a lung nodule dataset, along with a deep learning approach for image analysis and nodule classification.	JSRT PLCO
2	29	[42,43]	CT/CXR	Connected using local feature analysis, linear filtering, clustering techniques, and other non-deep learning techniques.	LIDC
3	24	[44]	CXR	Artificial intelligence and machine learning methods, including ANN, SVM, and KNN. Typically used the JSRT database.	JSRT
4	23	[45]	CXR	Rib/bone suppression and image enhancement techniques, including wavelet transform methods.	JSRT
5	17	[46,47]	CT	Use of deep learning and shape analysis to diagnose lung cancer from chest CT images.	Luna16
6	12	[48,49]	CXR	KNN classification of nodules as blobs. Used stratification of JSRT to train/calibrate schemes to reduce false positive detection by algorithms.	JSRT
7	12	[50]	CXR	A set of older papers using various techniques to detect nodules and reduce false-positive detections	Private Data JSRT

**Table 2 sensors-23-06585-t002:** Summary of the datasets used in this study.

Dataset	Nodule Image Count	Non-Nodule Image Count	Image Size/Format	Label Accuracy AUC-ROC
JSRT	154 images from 154 patients	93 images from 93 patients	Universal Image Format 2048 × 2048 12-bit grayscale	20 radiologists from 4 institutions. 0.833 ± 0.045
LIDC	280 images from 157 patients	0	DICOM Extracted and compressed to 512 × 512 PNG using Pydicom [53]	Not provided

**Table 3 sensors-23-06585-t003:** Experiment/ablation tests covering each element of the proposed debiasing pipeline.

Experiment	Segmentation	Cropping	Rib Suppression	Sample Image
A	False	False	False	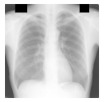
B	False	False	True	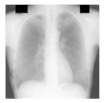
C	True	False	False	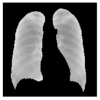
D	True	False	True	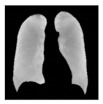
E	True	True	False	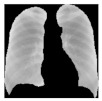
F	True	True	True	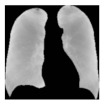

**Table 4 sensors-23-06585-t004:** Top five most difficult JSRT nodule images pruned for experiment E. Pruning these images resulted in a stable AUC of around 78%. A circle represents nodule location per JSRT metadata. All nodules are in challenging positions even for human radiologists, except for JPCLN142.png, which is at the threshold of human visibility.

Filename	Image	JSRT Metadata Notes	Radiologist Observations
JPCLN151.png	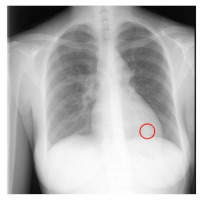	Extremely subtle 14 mm	Extremely subtle Behind cardiac silhouette Overlaps vascular marking
JPCLN003.png	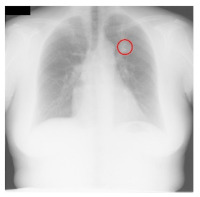	Obvious 30 mm	Obvious Overlaps vascular markings
JPCLN130.png	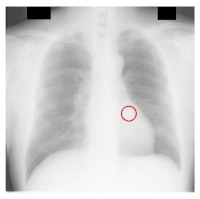	Extremely subtle 30 mm	Extremely subtle Behind cardiac silhouette
JPCLN141.png	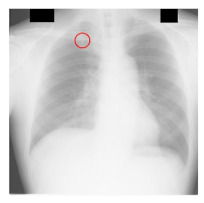	Extremely subtle 10 mm	Extremely subtle Behind rib/clavicle
JPCLN142.png	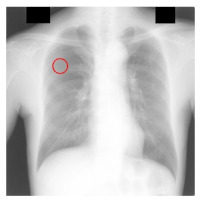	Extremely subtle 10 mm	Not visible.

## Data Availability

Publicly available datasets were analyzed in this study. These data can be found at the cited locations.

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
