# Peer review of "Development of Debiasing Technique for Lung Nodule Chest X-ray Datasets to Generalize Deep Learning Models"

_sensors, 2023, doi:10.3390/s23146585_

Round 1

Reviewer 1 Report

Thank you for giving me the opportunity to review this interesting research.

I believe it is wonderful that screening can be done using CXR considering the significant differences in radiation exposure and cost compared to CT.

Although I am aware of the limitations of CXR examinations, I think there are some problems in overcoming the current problems in this research.

Regarding areas that are prone to oversight during screenings, attention should be given to the hilum, diaphragm, heart, lung apices, and overlapping with the ribs. 

How about the effects on organs other than the ribs?

Cases where there are pacemakers, post-lung resection with displacement, or significant skeletal deformities such as kyphosis can pose challenges in screenings.

How about addressing these points?

How about the detection of abnormal findings related to other conditions like mediastinal tumors?

Author Response

Thank you for the kind feedback. Please see attached.

Reviewer 2 Report

The authors did a thorough investigation on the development of a debiasing deep leaning models for lung nodule chest X-ray analysis. This article discussed the application of rib/bone suppression operator in this model and how it helps improve the accuracy of the studied model.  Overall, the manuscript did a comprehensive investigation regarding the deep leaning model development and will attract wide interest in this field. I suggest accept in present form. The only comment I have for the authors is to consolidate some of the figures, Fig 9-14 for example and may put some figures in the supplementary documents.

Author Response

Reviewer 2

The authors did a thorough investigation on the development of a debiasing deep leaning models for lung nodule chest X-ray analysis. This article discussed the application of rib/bone suppression operator in this model and how it helps improve the accuracy of the studied model.  Overall, the manuscript did a comprehensive investigation regarding the deep leaning model development and will attract wide interest in this field. I suggest accept in present form. The only comment I have for the authors is to consolidate some of the figures, Fig 9-14 for example and may put some figures in the supplementary documents.

Author Response:

Thank you again for this excellent suggestion and your constructive comments, which we greatly appreciate. This version of the manuscript does consolidate Figures 9-14 into figure 9 resulting in a much more readable and understandable discussion. The authors believe that this change has made the document less cumbersome and improved its overall flow. Therefore we are hesitant to extract figures and move them to supplementary materials, as we prefer to maintain all within the main manuscript.

Round 2

Reviewer 1 Report

The manuscript was appropriately revised, so it's worth accepting it.